# Life Prediction of Battery Using a Neural Gaussian Process with Early Discharge Characteristics

**DOI:** 10.3390/s21041087

**Published:** 2021-02-05

**Authors:** Aijun Yin, Zhibin Tan, Jian Tan

**Affiliations:** 1State Key Laboratory of Mechanical Transmissions, Chongqing University, Chongqing 400044, China; zhibin.tan@cqu.edu.cn; 2College of Mechanical Engineering, Chongqing University, Chongqing 400044, China; 3Chongqing Gas Field, PetroChina Southwest Oil and Gas Field Company, Chongqing 400021, China; tanjian312897@163.com

**Keywords:** batteries, life prediction, neural Gaussian process, early discharge characteristics

## Abstract

The state of health (SOH) prediction of lithium-ion batteries (LIBs) is of crucial importance for the normal operation of the battery system. In this paper, a new method for cycle life and full life cycle capacity prediction is proposed, which combines the early discharge characteristics with the neural Gaussian process (NGP) model. The cycle data sets of commercial LiFePO_4_(LFP)/graphite cells generated under different operating conditions are analyzed, and the power characteristic *P* is extracted from the voltage and current curves of the early cycles. A Pearson correlation analysis shows that there is a strong correlation between *P* and cycle life. Our model achieves 8.8% test error for predicting cycle life using degradation data for the 20th to 110th cycles. Based on the predicted cycle life, capacity degradation curves for the whole life cycle of the cells are predicted. In addition, the NGP method, combined with power characteristics, is compared with other classical methods for predicting the remaining useful life (RUL) of LIBs. The results demonstrate that the proposed prediction method of cycle life and capacity has better battery life and capacity prediction. This work highlights the use of early discharge characteristics to predict battery performance, and shows the application prospect in accelerating the development of electrode materials and optimizing battery management systems (BMS).

## 1. Introduction

In recent years, lithium-ion batteries (LIBs) have attracted widespread attention due to the advantages of their high energy density, low self-discharge characteristics, and absence of memory effects [1,2,3]. They have become the preferred energy storage systems for many engineering and industrial applications, such as portable devices, automobiles, and aerospace systems [4,5,6]. However, the performance of LIBs will deteriorate with the decrease of capacity, which may lead to equipment and system degraded capability, failures, or even catastrophic loss [7].

State of health (SOH) is an index used to evaluate the aging degree of cells, which includes capacity fade and cycle life prediction. It provides very useful information to predict when a battery should be replaced. Battery performance is usually monitored by measuring battery voltage, current, temperature, capacity, impedance, and other parameters through relevant experiments. Various corresponding methods can be used to evaluate the battery health status based on the parameters mentioned above [8]. Cycle life refers to the number of cycles in which available capacity decays to 80% of the rated capacity of the battery [9]. Therefore, it is vital to achieve an accurate and reliable prediction of SOH of LIBs, which is a key approach of battery management system (BMS) in scientific research and practical application [10]. Meanwhile, the accurate prediction model of capacity decay and cycle life statistics could significantly accelerate the development and commercialization of LIBs, such as accelerating the development of novel electrode materials with larger capacities and longer lives by material design and evaluating the cycle life of batteries [11,12]. 

Currently, model-based and data-driven methods are the most common methods used to predict cycle life and capacity degradation [13]. Model-based methods use mathematical models which are defined according to the physical degradation mechanism or reasonable experience of the battery, to capture the law of battery degradation [9]. Salkind et al. [14] used fuzzy logic mathematics to analyze data obtained by Electrochemical Impedance Spectroscopy (EIS) and coulomb counting techniques, developing a practical method for the state of charge (SOC) and SOH prediction of batteries. Eddahech et al. [15] predicted the battery life based a single parameter identified from EIS tests. The authors of [16] developed a battery nonlinearities model using circuit parameters such as resistors, capacitors, and inductors that were based on a modified Randles circuit model to predict the capacity fade. However, the accuracy of the EIS measurements is impacted by the noises caused by the other integrated components of an online system [17]. Equivalent circuit models with a large number of unknown parameters capture all key behaviors of battery cells, and are complex and diverse [18]. Therefore, on the basis of a reasonable battery capacity attenuation definition, many researchers presented life prediction models combined with advanced filtering technology. Xing et al. [17] proposed an empirical exponential and a polynomial regression model to track the degradation trend of cells over their cycle life based on experimental data analysis, and used a particle filtering approach to adjust model parameters online. In Ref. [19], a novel model was developed using an unscented Kalman filter with relevance vector regression, and applied to the cycle life and short-term capacity prediction of batteries. Su et al. [20] developed a new prognostic method for determining battery cycle life based on the interacting multiple model particle filter (IMMPF). Their method, that applies the IMMPF to different state equations, was used for multiple capacity models of LIBs. Model-based methods have made substantial progress, and their performance has been verified by different experiments; however, the accuracy and robustness of those model are limited by the accuracy of the battery degradation physical model [21].

Simultaneously, cycle life prediction based on data-driven methods has also been widely studied. The data-driven methods can directly capture the degradation evolution law of LIBs without using a complex mathematical model to define the degradation mechanisms of LIBs [22], which makes cycle life prediction methods based on statistical and machine learning techniques attractive. The authors of [22] put forward a Naive Bayes model to predict the cycle life of cells under different operating conditions. In Ref. [23], the correlation vector derived from the relevance vector machine (RVM), which can reflect the capacity degradation trajectory, was used to predict the cycle life of LIBs. Instead of using capacity to predict SOH, many researchers have constructed battery health indicators (HIs) to improve estimation accuracy and efficiency, such as equal interval discharge voltage difference [24], average voltage attenuation [25], and discharge voltage sampling entropy [26]. Patil et al. [27] presented a real-time cycle life estimation method based on a support vector machine. In this method, the key features were extracted from the voltage and temperature distribution, and the cycle data of LIBs under different operating conditions were analyzed. Liu et al. [24] used HIs, which were extracted based on the charging and discharging voltage, current, and temperature, to analyze the degradation of online LIBs. The accuracy and stability of the proposed method was further improved by using an optimized RVM algorithm. In Ref. [21], the SOC was estimated using a Gaussian process (GP) regression framework and HIs established by voltage, current, and temperature. This method obtained the prediction probability distribution of SOC, which removed the restrictions of the previous methods, in which they could only give the point estimation of SOC. In Ref. [28], a prediction model combining parameter optimization and method hybrid Gaussian process function regression was proposed to achieve high precision predictions. In addition to HIs, neural network techniques have also been used to predict the SOH of lithium-ion batteries. Wu et al. [29] used the battery terminal voltage curve to describe the remaining useful life (RUL) of LIBs, and proposed a LIBs cycle life estimation method based on a feedforward neural network and the importance sampling. In Ref. [30], a deep neural network was used to predict the SOH and RUL of batteries. In Ref. [31], a SOC estimation method was introduced through a combination of a radial basis function neural network (RBFNN), an orthogonal least-squares algorithm, and an adaptive genetic algorithm. The method was validated using LiFePO_4_ (LFP) batteries under several different discharging conditions. Currently, there are few reports on the use of early data to predict cycle life. Recently, Severson et al. [32] proposed a prediction model combining data generation and a data-driven model, which used the discharge voltage curve of the early cycles to predict and classify the cycle life of LIBs. Their research showed the prospect of accurately predicting battery life using early discharge characteristics.

In this study, we explore the possibility of extracting battery health indicators from early discharge characteristics to predict SOH. The proposed methodology is combined with the Neural Gaussian Process (NGP) model [33,34,35], which considers battery degradation under various operating conditions. First, based on a correlation analysis, we predicted that the average power that decays in the early cycles of batteries is related to the cycle life. Afterwards, the cycle life of LIBs is predicted using the established model. The proposed method is then compared with other cycle life and capacity prediction methods.

The rest of the paper is organized as follows. The battery data preparation is presented in Section 2. The physical feature extraction is shown in Section 3. In Section 4, the methodology is introduced. The results and description are discussed in Section 5. Finally, the conclusion is summarized in Section 6.

## 2. Battery Data Preparation

We used the degradation data set of 124 commercial LFP/graphite cells (A123 Systems, model APR18650-M1A, 1.1 Ah nominal capacity, failure capacity of 0.88 Ah) generated by Severson et al. [32]. In the experiment, those cells were cycled in a forced convection temperature chamber set to 30 °C, and the detection parameter data sets were recorded by the Arbin LBT potentiostat sensor. Parameters such as voltage, capacity, current, temperature, and internal resistance were continuously measured during the cycles. The data set was generated by three batches of batteries through tests, of which the first batch contained 41 batteries, the second batch contained 43 batteries, and the third batch contained 40 batteries. All cells in this dataset were charged with a two-step fast-charging policy. This is the largest publicly available dataset for nominally the same commercial LIBs cycled under controlled conditions. We studied the performance decline of LIBs with multiple degradation mechanisms due to the manufacturing processes and operating conditions of the dataset. Meanwhile, fast charging conditions made it possible to estimate battery health and optimize battery performance under extremely fast conditions, and accelerate the commercialization of fast-charging lithium batteries.

Figure 1 shows the functional relationship between the observed discharge capacity and the number of cycles in the whole life cycle. The capacity of the batteries decay slowly during the early cycles, while rapidly decaying in the later stage of the cycles. The degradation tracks of the batteries are interlaced with each other. This indicates that there is a complex, nonlinear relationship between capacity and cycle number, and it is difficult to choose an accurate aging model as the basis of battery dataset life prediction [36]. In order to overcome these difficulties, we combined key physical characteristics, which were constructed by the early discharge characteristics and reflected the state of health of LIBs, with a data-driven approach to predict the cycle life of cells.

## 3. Physical Feature Extraction

In previous reports, researchers usually used measurement parameters such as voltage, current, and temperature to construct physical properties to predict battery life [24,27,29]. In this paper, in order to explore the relationship between early discharge characteristics and cycle life, we use discharge voltage U(t), discharge capacity Q(t), and discharge time t in every cycle to propose the average power characteristic P, which is a function of cycle number. The power characteristic $P_{jk}$ of a battery per cycle is defined as
(1)$$P_{jk}=\sum_{i=2}^{n}U\left(t_{i}\right)\left(Q\left(t_{i}\right)-Q(t_{i-1}\right))/\left(t_{n}-t_{1}\right)$$
where *j* represents the *j*th battery, and *k* stands for the kth cycle. For the kth cycle of the *j*th cell, $U\left(t_{i}\right)$ is the discharge voltage of the cells at time $t_{i}$, where *i* is a moment in a discharge cycle i=2,3,…,n. $Q\left(t_{i}\right)$ and $Q(t_{i-1})$ are respectively the discharge capacity of the batteries at time $t_{i}$ and time $t_{i-1}$ in the cycle. Figure 2a is P attenuation curve of the cells in the early cycles. Since the capacity increased slightly in the early discharge stage, in order to study the relationship between the characteristics of the decay phase and cycle life, we use data from the 20th to 110th cycles. P shows a fluctuating attenuation trend, and its attenuation is very small from the whole life cycle. We use the variance statistics to convert the fluctuation attenuation of the average power characteristic $P_{j}$ of each battery from the 20th to 110th cycles into a scalar $P_{Dj}$ to establish a corresponding relationship with the cycle life $L_{j}$. The relationship between $P_{D}$ and cycle life L is quantitatively analyzed by the Pearson rank correlation coefficients. It is expressed as,
(2)$$\rho_{P_{D},L}=\frac{E\left(P_{D}L\right)-E\left(P_{D}\right)E\left(L\right)}{\sqrt{E\left(P_{D}^{2}\right)-{\left(E\left(P_{D}\right)\right)}^{2}}\sqrt{E\left(L^{2}\right)-{\left(E\left(L\right)\right)}^{2}}}$$

Surprisingly, Figure 2b shows a significant correlation between the variance of characteristic *P* and cycle life of LIBs ($\rho_{P_{D},L}=-0.936$). Compared with previous studies, the correlation coefficient of power characteristic is higher than that of the voltage characteristic using the 110th cycle and the 20th cycle ($\rho_{\Delta Q_{110-20}\left(V\right)}=-0.932$, where capacity Q is a function of voltage, $\mathrm{and}\;\Delta Q_{110-20}\left(V\right)=Q_{110}\left(V\right)-Q_{20}\left(V\right)$) [32].

## 4. Methodology

### 4.1. Neural Gaussian Process Model

Although P is strongly correlated with the cycle life when considering the nonlinear relationship between cycle number and the discharge capacity, the linear model fails to predict the capacity decay. Therefore, we built a prediction model based on characteristic P and NGP.

NGP is a machine learning method that combines neural network and GP. It makes use of the advantages of the neural network function approximation to solve the difficulty of selecting the kernel function of Gaussian process through experience. This allows the model flexibility to carry out nonlinear simulation to achieve target prediction.

Figure 3 illustrates the schematic structure of the NGP. In order to learn and optimize the parameters of the model and generate target prediction, the model divides the data set $\left(x_{1:n},y_{1:n}\right)$ into a training set $(x_{1:m},y_{1:m}),\;1<m<n$ and test set $\left(x_{m:n},y_{m:n}\right)$, where $x_{1:n}≔\left(x_{1},\ldots ,x_{n}\right),y_{1:n}≔\left(y_{1},\ldots ,y_{n}\right)$. In particular, in the training and test stages of the model, the data consists of the context set $\left(x_{C},y_{C}\right)$ and the target set $\left(x_{T},y_{T}\right)$. In the training phase, the context set $\left(x_{C},y_{C}\right)$ and the target set $\left(x_{T},y_{T}\right)$ are derived from the training set $\left(x_{1:m},y_{1:m}\right)$. In the test phase, the context set $\left(x_{C},y_{C}\right)$ is the training set $\left(x_{1:m},y_{1:m}\right)$ and the target set $\left(x_{T},y_{T}\right)$ is the entire data set $\left(x_{1:n},y_{1:n}\right)$. The marginal joint distribution of the objective function $F\left(x_{T}\right)$ is defined as,
(3)$$p\left({\hat{y}}_{T}|x_{T},x_{C},y_{C}\right)=\int p\left({\hat{y}}_{T}|x_{T},k_{*},z_{C}\right)p(z_{C}|s_{C})dz_{C}$$
where ${\hat{y}}_{T}$ is the prediction target vector. *p* denotes the abstract probability distribution over all random quantities, z is the latent variable of the model, obeys the Gaussian distribution, and measures the probability uncertainty of the model. $s_{C}$ is a parameter generated by the context set $\left(x_{C},y_{C}\right)$, and $z_{C}\sim N\left(\mu \left(s_{C}\right),I\sigma \left(s_{C}\right)\right)$ is modelled by a Gaussian parameter by $s_{C}$. $k_{*}$ is the attention mechanism of the model. This mechanism makes full use of the information of $x_{C}$, and gives important contextual points that have more weight within the model. The mechanism of attention can be defined by the Laplace kernel, the dot-product kernel, etc. At the same time, the parameters of the encoder and decoder are learned by maximizing the following evidence lower bound (ELBO),
(4)$$\log p\left(y_{T}^{'}\text{|}x_{T},k_{*},x_{C},y_{C}\right)\geq E_{q(z|x_{T},y_{T})}\left[\log p\left(y_{T}\text{|}z_{T},k_{*},x_{T}\right)-\text{log}\left(q(z_{T}|s_{T})/q(z_{C}|s_{C})\right)\right]$$
where $q(z_{T}|s_{T})$ and $q(z_{C}|s_{C})$ are the variational posteriori of $p(z_{T}|s_{T})$ and $p(z_{C}|s_{C})$, respectively. During the training, the target values $y_{T}$ are known, and Equation (4) is used to optimize the encoder and decoder parameters. And $y_{T}^{'}$ is the prediction vector of *x_T_*. During the test, Equation (3) is used to predict the target values ${\hat{y}}_{T}$. 

### 4.2. Prediction Framework

The procedure of the battery cycle life prediction, based on early discharge characteristics, is presented in Figure 4. Moreover, the methods of voltage characteristic and the RBFNN model are compared, respectively, to illustrate the advantages of the power characteristic and NGP model proposed in this paper.

Suppose the battery data is $\left(x_{1:n},L_{1:n}\right)$, and divide the data set into training set $(x_{1:m},L_{1:m}),1<m<n$ and test set $\left(x_{m:n},L_{m:n}\right)$. The prediction step of battery life can be expressed as follows: firstly, the power characteristic $P_{D}$ of all cells is calculated by Equation (1). Then, the battery life $\hat{L}$ is predicted by Equations (3) and (4). At the same time, according to the results of the single point life prediction, we use the same data as the life prediction to realize the prediction of the whole life cycle capacity degradation of cells. Specifically, we normalize $C_{y1:\hat{L}}≔C_{y1},\ldots ,C_{y\hat{L}}$, denoted as $C_{y1:\hat{L}}^{*}$, and establish a functional relationship with $Q_{1:\hat{L}}≔Q_{1},\ldots ,Q_{\hat{L}}$. Normalization is defined as
(5)$$C_{y1:\hat{L}}^{*}=C_{y1:\hat{L}}/\hat{L}$$

Similar to the battery cycle life prediction, the training set data is used to optimize the model. For the test batteries, the discharge capacity data $\left(C_{y1:110}^{*},Q_{1:110}\right)$ of the first 110 cycles are inputted into the model to obtain the capacity $\left(C_{y1:\hat{L}}^{*},Q_{1:\hat{L}}\right)$ of the entire life cycle.

## 5. Results and Description

### 5.1. Cycle Life Prediction

The battery data set consists of three sampling batches; we divided the data into a training set, the first test set, and the second test set to verify the predictive performance and adaptability of the model. In particular, we combined the first two batches of data and divided the data set into the training set (41 cells) and the first test set (43 cells, by intermittent sampling, and divided the battery data of the third batch into the second test set (40 cells). To evaluate the prediction accuracy of the proposed method, the mean absolute percent error (MAPE) is used in this paper. It is given by
(6)$$\mathrm{MAPE}=\frac{1}{n}\overset{n}{\underset{j=1}{\sum }}\left|L_{j}-{\hat{L}}_{j}\right|/L_{j}\;\times 100\%$$
where $L_{j}$ is the observed cycle life of the *j*th battery, ${\hat{L}}_{j\;}$ is the model predicted life of the *j*th battery, and *n* represents the number of training sets or test sets.

In this paper, the validity of the proposed power characteristic and model is illustrated by comparing the cycle life prediction performance of the voltage characteristic and the RBFNN model. Figure 5 and Figure 6 show the predicted performance of the power characteristics using the NGP model and the RBFNN model, respectively. Figure 7 and Figure 8 compare the predicted results of the voltage characteristics using the NGP model and the RBFNN model. A comparison between the predicted cycle life and the observed cycle life is shown in Figure 5a, Figure 6a, Figure 7a and Figure 8a. Table 1 displays the MAPE of cycle life for each method. A smaller MAPE value indicates a better data set cycle life prediction performance. Surprisingly, the predicted error of the NGP model, using power characteristics, is 12.0% in the first test set and 8.8% in the second test set, using only the 20th to 110th cycles data. It is worth noting that the battery performance has only a very weak decay during the first 110 cycles. In the first test set, excluding a rapidly degraded battery that deviates from the relationship between *P* and the cycle life of the other batteries, the predicted error is reduced to 8.8%. Simultaneously, the prediction errors of the RBFNN model using power characteristics in the training set, the first test set, and the second test set are 10.2%, 10.1%, and 9.8%, respectively. The prediction errors of the NGP model using voltage characteristics in the training set, the first test set, and the second test set are 9.3%, 10.7%, and 11.9%, respectively. The prediction errors of the RBFNN model using voltage characteristics in the training set, the first test set, and the second test set are 10.5%, 12.5%, and 12.6%, respectively. As can be seen from Table 1, NGP performs better than RBFNN when using the same battery characteristics. At the same time, using the same model, the predictive performance of the power characteristics optimizes the voltage characteristics. The prediction curves of each method are shown in Figure 5b, Figure 6b, Figure 7b and Figure 8b. For voltage characteristics, the prediction curve of the NGP model is smoother than that of the RBFNN model. The RBFNN model shows a sharp value in the lower left corner of Figure 8b. For the power characteristics, the prediction curves of the two models are smooth. The results show that the proposed power characteristics can improve the prediction accuracy of the model, and the NGP model has a better regression fitting performance. The NGP model, combined with the power characteristics, is an effective method to predict the cycle life of batteries.

### 5.2. Capacity Prediction

In this brief, batteries with a different cycle life (i.e., cycle life 335, cycle life 854, cycle life 1028, and cycle life 1638) are selected to verify the NGP model capacity prediction results, and compare the prediction results of the RBFNN model. As the prediction results of the RBFNN model are affected by the length of the training data, we not only use the data of the first 110 cycles to predict capacity, but also use the data of the first 20% of cycles of the predicted life of the power characteristics to predict capacity. The MAPE of the cycle that reaches the capacity failure threshold for the first time is used to quantify the performance of the method, based on NGP and RBFNN.

Figure 9 shows the capacity prediction results of the NGP and RBFNN models. The details of the capacity prediction results can be seen in Table 2 and Table 3. Figure 9 shows that the capacity prediction curve of the NGP model fits the real curve better. For the failure life predicted by the capacity curve, the prediction error of the NGP model is less than 5% using the first 110 cycle data, and the performance growth is small after changing the length of training, while the performance of the RBFNN model, using only the data of the first 110 cycles, is poor with the increase of battery cycle life. When using the data of the first 20% cycles of the predicted life of the power characteristics, the RBFNN model shows better performance, but is still worse than the NGP.

The results show that the data length has little effect on the performance of the NGP model, but has a great influence on the RBFNN model. In practice, the battery discharge data are measured, so it is difficult to increase the training length of the model. Combined with the prediction results of cycle life and capacity, the NGP model shows good prediction accuracy.

## 6. Conclusions

This paper proposes a battery life prediction model, combining discharge characteristics and NGP, using early cycle discharge data. For the degradation data generated by commercial LFP/graphite batteries under different cycling conditions, we do not use a priori battery degradation knowledge. Rather, on the one hand, we extract the average power characteristic from the discharge voltage, battery capacity, and charging time, and, on the other hand, combined with NGP, using only the 20th to 110th cycle data, we get 8.8% life predicted error and a complete capacity degradation predicted curve. The prediction results show that the performance of the NGP and power characteristic model is better than that of the traditional FRBFF and voltage characteristic techniques. Thus, the proposed method NGP with early discharge characteristics has superior performance in cycle life and capacity prediction. In general, our approach can optimize material design, and provides a reliability assessment for the BMS by evaluating battery performance early in the battery cycle. In future, new studies should consider more battery discharge characteristics in order to improve prediction performance.

## Figures and Tables

**Figure 1 sensors-21-01087-f001:**
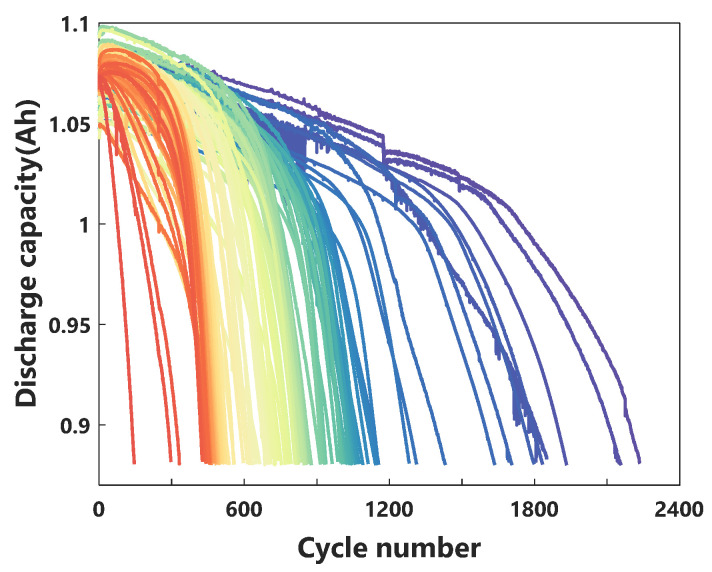
The observed discharge capacity curves; the color of the curves changes along the spectrum according to the cycle life.

**Figure 2 sensors-21-01087-f002:**
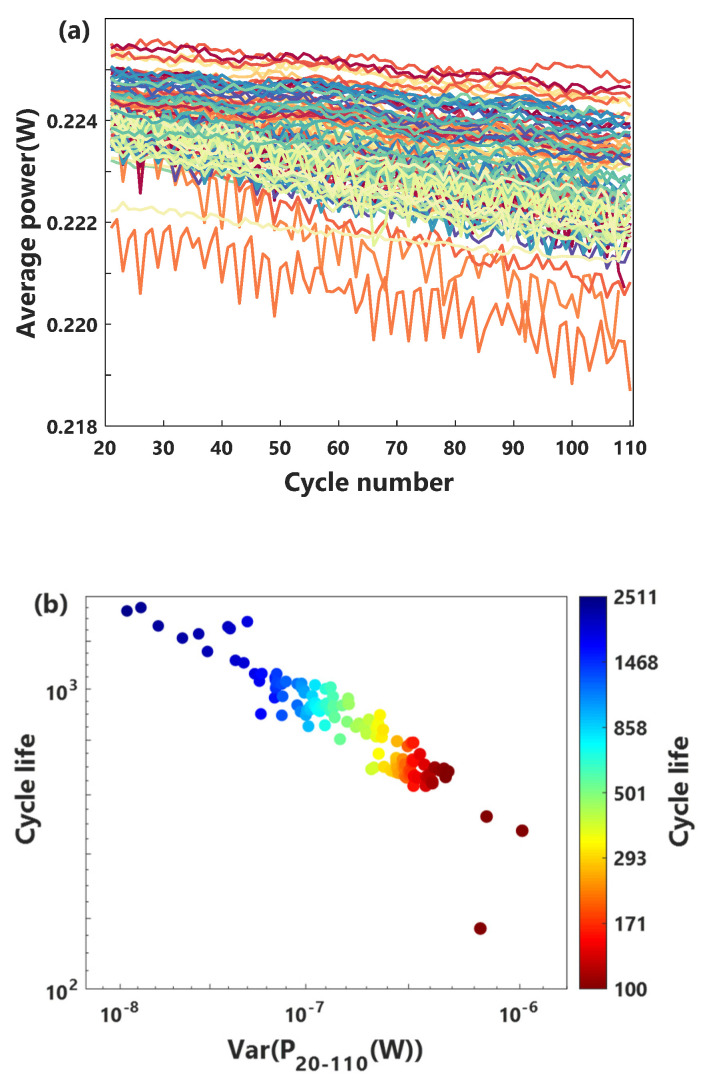
(**a**) *P* decayed slowly during the 20th to 110th cycles, in which the abrupt data caused by the recording error is deleted, and (**b**) the correlation analysis of the logarithm of the variance of *P* and the logarithm of the cycle life.

**Figure 3 sensors-21-01087-f003:**
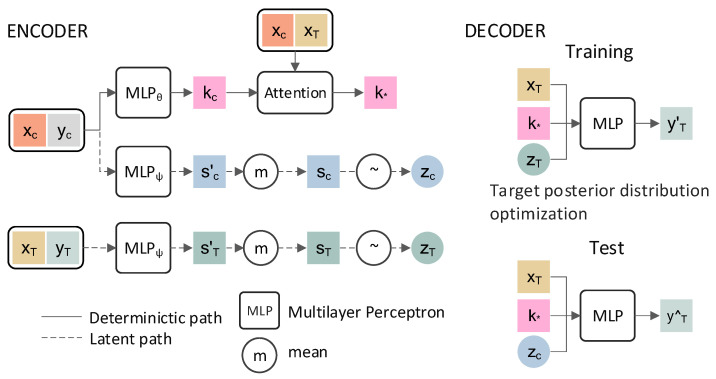
NGP model framework. The calculation processes are defined by MLP, except for the attention mechanism and mean. The deterministic path generates attention mechanism $k_{*}$, and the latent path generates latent variable $z_{C}$ and $z_{T}$. $k_{C},s_{C}^{\prime }$ and $s_{T}^{\prime }$ are the intermediate variables that generate $k_{*},\;z_{C}$, and $z_{T}$, respectively. The methods defined by $k_{C},s_{C}^{\prime }$ and $s_{T}^{\prime }$ are similar. For instance, $k_{C}$ is a set composed of $k_{Ci}$, which is a representation produced by each pair ${\left(x_{C},y_{C}\right)}_{i}$ in the input space through MLP.

**Figure 4 sensors-21-01087-f004:**
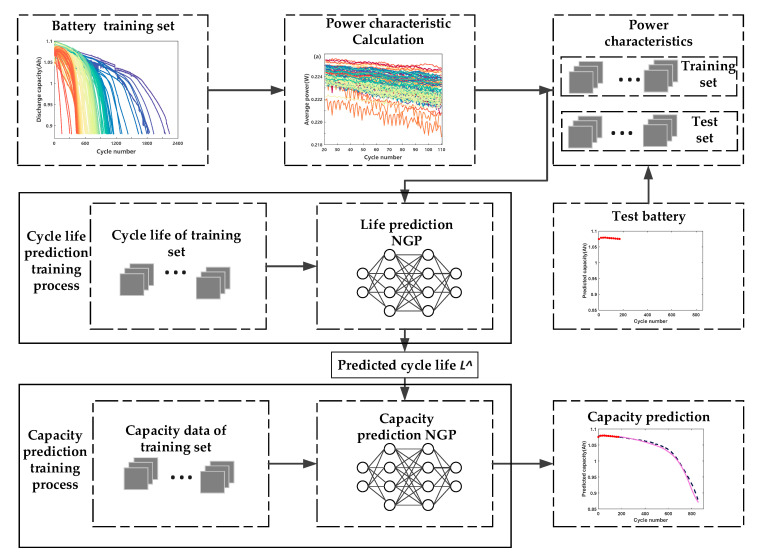
The flow chart of battery cycle life and capacity prediction.

**Figure 5 sensors-21-01087-f005:**
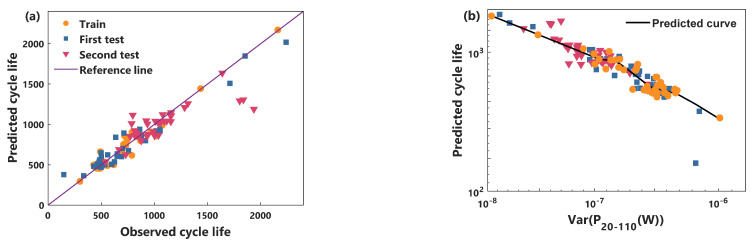
NGP and power characteristic model cycle life prediction. (**a**) Comparison of observed life and predicted life, (**b**) the cycle life predicted curve.

**Figure 6 sensors-21-01087-f006:**
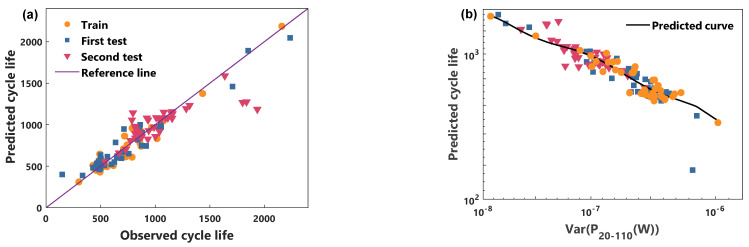
RBFNN and power characteristic model cycle life prediction. (**a**) Comparison of observed life and predicted life, (**b**) the cycle life predicted curve.

**Figure 7 sensors-21-01087-f007:**
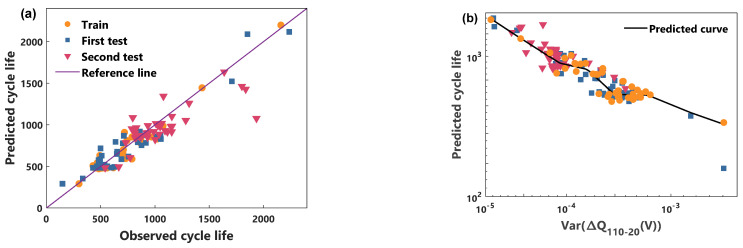
NGP and voltage characteristic model cycle life prediction. (**a**) Comparison of observed life and predicted life, (**b**) the cycle life predicted curve.

**Figure 8 sensors-21-01087-f008:**
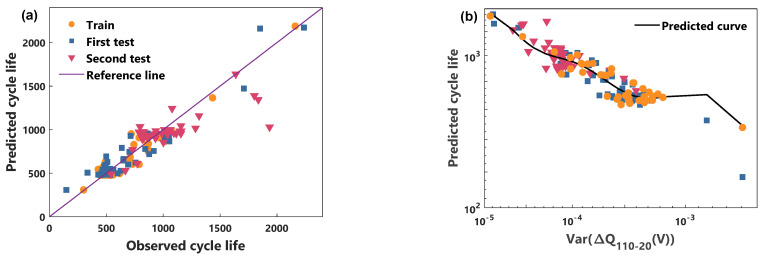
RBFNN and voltage characteristic model cycle life prediction. (**a**) Comparison of observed life and predicted life, (**b**) the cycle life predicted curve.

**Figure 9 sensors-21-01087-f009:**
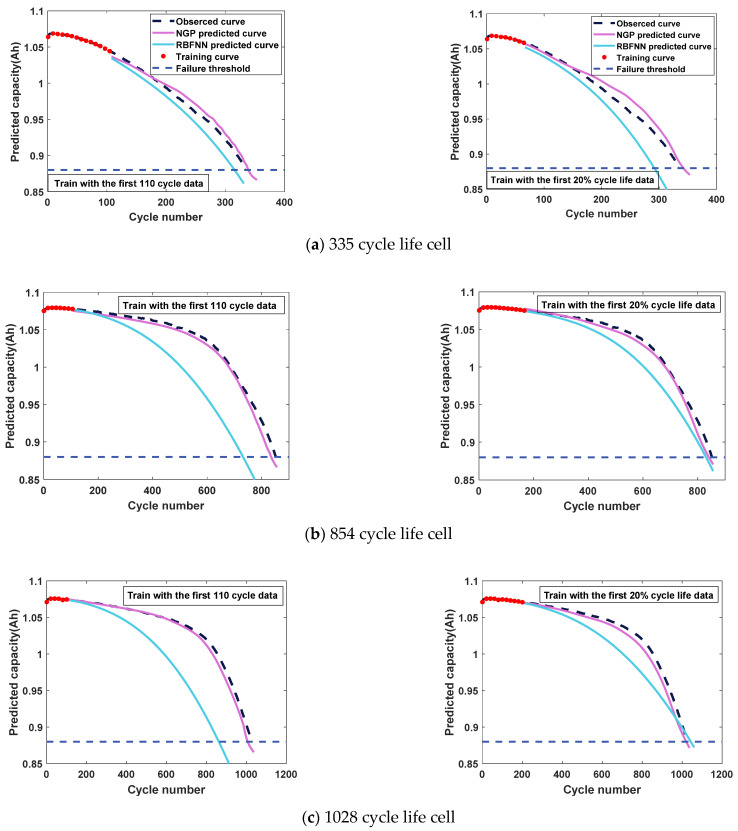
The predicted discharge capacity curves.

**Table 1 sensors-21-01087-t001:** Comparison of battery cycle life prediction performance.

Data Set	MAPE of Model with Characteristics			
	Power with NGP (%)	Power with RBFNN (%)	Voltage with NGP (%)	Voltage with RBFNN (%)
Train	8.4	10.2	9.3	10.5
First test	12.0 (8.8)	13.8 (10.1)	12.7 (10.7)	14.7 (12.5)
Second test	8.8	9.8	11.9	12.6

The results in parentheses exclude a battery that deviates from the relationship between *P* and cycle life, or between *V* and cycle life.

**Table 2 sensors-21-01087-t002:** Failure cycle life prediction based on NGP and RBFNN models.

Model	Cycle Life 335	Cycle Life 854	Cycle Life 1028	Cycle Life 1638
Power *P* predicted cycle life	353	856	1037	1638
NGP predicted failure cycle (110 cycles)	339	834	1006	1581
NGP predicted failure cycle (20% predicted of cycle life)	342	840	1020	1588
RBFNN predicted failure cycle (110 cycles)	315	731	861	1828
RBFNN predicted failure cycle (20% predicted of cycle life)	291	830	1043	1712

**Table 3 sensors-21-01087-t003:** MAPE comparison of failure cycle predicted by capacity curve of NGP and RBFNN models.

Model	Cycle Life 335 (%)	Cycle Life 854 (%)	Cycle Life 1028 (%)	Cycle Life 1638 (%)
NGP with 110 cycles	1.2	2.3	2.1	3.5
NGP with 20% predicted of cycle life	2.1	1.6	0.8	3.1
RBFNN with 110 cycles	6.0	14.4	16.2	11.6
RBFNN with 20% predicted of cycle life	13.1	2.8	1.4	4.5

## Data Availability

The data that support the findings of this study are available from the corresponding author upon reasonable request.

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
