# Peer review of "Life Prediction of Battery Using a Neural Gaussian Process with Early Discharge Characteristics"

_sensors, 2021, doi:10.3390/s21041087_

Round 1
Reviewer 1 Report
The manuscript presents well-prepared research work on commercial lithium iron phosphate batteries. The aim of the work is to predict battery cycle life using a Neural Gaussian Process on the early discharge stage. The general structure of the work is well organized, it is easy to read and reach the Authors' impact in the field.
Comments by sections:
- The title is informative and relevant. The abstract matches the rest of the article.
- The introduction section is well organized and present a proper background study, it is clear what is already known about this topic. The research question is clearly outlined and has a straight focus.
- The way of observing batteries' cycle life data is correct, and physical features extraction appropriately done. Methods used in research work seems to be valid and relabel. The variables are defined and measured appropriately.
- The methodology of Neural Gaussian Process training is well described and clear for the reader.
- Results are well presented. Tables and figures are relevant and well described in the text. It is clear what is a practically meaningful result of the research work.
- Conclusions are supported by results and answer the aims of the study.
Specific formatting comments:
- It seems that the LFP state for lithium iron phosphate battery, however, is not defined in the text (neither in the abstract). Please keep in mind that abbreviations and acronyms must be defined the first time they are used in the text, even after they have been defined in the abstract.
- "Gaussian process" is mentioned for the first time in line 84, but abbreviation defined in line 100, later once again in 104 and in 167
- Abbreviation RUL (line 89) is not defined, at the same time, several abbreviations are given but not used in the text after the introduction section (e.g., RVM, IS, UKF, RVR, etc.)
- Line 143: variable kth is presented as regular text instead of being presented as a part of the equation.
- Description of Neural Gaussian Process in the methodology section (Lines 167-170) repeats the introduction section.
- Please add units (at least %) to Table 1
- Table 1: in the first column, "second test" must be capitalized.
Author Response
Thanks for the reviewers’ comments concerning our manuscript entitled “Life prediction of battery using a neural Gaussian Process with early discharge characteristics” (sensors- 105793). Those comments are all valuable and very helpful for revising and improving our paper, as well as the important guiding significance to our researches. We have studied comments carefully and have made correction which we hope meet with approval. Revised portion are highlighted in the paper.

Reviewer 2 Report
This paper employed a neural Gaussian process model to estimate the battery life cycle. This paper has an average quality and can be improved further. Some of the major concerns are:
- The presented results show an 8.8% prediction error, which seems to be relatively high.
- The comparison used for the proposed method is not appropriate.
Further improvements.
- Article type “Communication.” Check if appropriate. This article sounds more like a research article.
- LFP and GP are not defined. Variable P is not recommended; consider omitting it.
- Some recent key articles from MDPI are missing. There are many good data-driven/gaussian process in battery application, such as:
- Peng, Y., Hou, Y., Song, Y., Pang, J., & Liu, D. (2018). Lithium-ion battery prognostics with hybrid Gaussian process function regression. Energies, 11(6), 1420.
- Khumprom, P., & Yodo, N. (2019). A data-driven predictive prognostic model for lithium-ion batteries based on a deep learning algorithm. Energies, 12(4), 660.
- Jia, J., Liang, J., Shi, Y., Wen, J., Pang, X., & Zeng, J. (2020). SOH and RUL Prediction of Lithium-Ion Batteries Based on Gaussian Process Regression with Indirect Health Indicators. Energies, 13(2), 375.
- Line 110. LFP is not defined; consider removing it.
- Line 143. There should be no indentation on where it should be a continuation from the previous paragraph. Check the rest of the paper.
- Line 144. Equation 1. i and n variable is not explained. “at time ti, where i is … i= 2,3,…,n.”
- Line 225, is the I in Eq. 6 the same as Eq 1? If not, consider using another variable. I”th” should be superscripted instead of subscripted.
- Section 4. Please provide a flow chart of the overall proposed methodology from the data input used up to prediction.
- Table 1. This is not an apple-to-apple comparison of prediction error. The authors should compare all methods on either power or voltage or both, but not comparing 2 methods in power and 1 method in voltage. Line 240-246 should be deleted or rephrased. If the authors would like to compare their proposed method with Stevenson et al., please employ the same measures, use the voltage characteristics.
- Consider comparing the proposed method with neural networks, other data-driven algorithms, or variants of the Gaussian process.
- In addition, prediction error should be identified in Table 1, not only the numerical numbers without any meaning, what do the numbers mean? The higher, the better? Or the lower, the better? Similarly, Table 2 can be arranged better.
- This paper is lack of discussion, what are the drawback, broader impacts, future work? All of these aspects are missing.
Author Response

(The authors gave the same response as above.)

Round 2
Reviewer 2 Report
Thank you for the response. The authors had significantly improved the paper and addressed the reviewer's comments properly. Please check the notation and abbreviation consistency throughout the paper. For example, Line 146, “For the kth cycle of. ”. The "the" is not superscripted.